# Effects of 12 Weeks *Cosmos caudatus* Supplement among Older Adults with Mild Cognitive Impairment: A Randomized, Double-Blind and Placebo-Controlled Trial

**DOI:** 10.3390/nu13020434

**Published:** 2021-01-29

**Authors:** Yee Xing You, Suzana Shahar, Nor Fadilah Rajab, Hasnah Haron, Hanis Mastura Yahya, Mazlyfarina Mohamad, Normah Che Din, Mohamad Yusof Maskat

**Affiliations:** 1Dietetics Programme and Centre for Healthy Aging and Wellness (H-Care), Faculty of Health Sciences, Universiti Kebangsaan Malaysia, Jalan Raja Muda Abdul Aziz, Kuala Lumpur 50300, Malaysia; yeexing@gmail.com; 2Biomedical Science Programme and Centre for Healthy Aging and Wellness (H-Care), Faculty of Health Sciences, Universiti Kebangsaan Malaysia, Jalan Raja Muda Abdul Aziz, Kuala Lumpur 50300, Malaysia; nfadilah@ukm.edu.my; 3Nutritional Sciences Programme and Centre for Healthy Aging and Wellness (H-Care), Faculty of Health Sciences, Universiti Kebangsaan Malaysia, Jalan Raja Muda Abdul Aziz, Kuala Lumpur 50300, Malaysia; hasnaharon@ukm.edu.my (H.H.); hanis.yahya@ukm.edu.my (H.M.Y.); 4Diagnostic Imaging and Radiotherapy Programme and Centre for Diagnostic, Therapeutic and Investigative Studies, Faculty of Health Sciences, Universiti Kebangsaan Malaysia, Jalan Raja Muda Abdul Aziz, Kuala Lumpur 50300, Malaysia; mazlyfarina@ukm.edu.my; 5Health Psychology Programme, Centre of Rehabilitation and Special Needs, Faculty of Health Sciences, Universiti Kebangsaan Malaysia, Jalan Raja Muda Abdul Aziz, Kuala Lumpur 50300, Malaysia; normahcd@ukm.edu.my; 6Department of Food Sciences, Faculty of Science and Technology, Universiti Kebangsaan Malaysia, UKM, Bangi 43600, Malaysia; yusofm@ukm.edu.my

**Keywords:** *Cosmos caudatus*, cognitive function, mood, biomarkers, flavonoids

## Abstract

*Cosmos caudatus* (CC) contains high flavonoids and might be beneficial in neuroprotection. It has the potential to prevent neurodegenerative diseases. Therefore, we aimed to investigate the effects of 12 weeks of *Cosmos caudatus* supplement on cognitive function, mood status, blood biochemical profiles and biomarkers among older adults with mild cognitive impairment (MCI) through a double-blind, placebo-controlled trial. The subjects were randomized into CC supplement (*n* = 24) and placebo group (*n* = 24). Each of them consumed one capsule of CC supplement (250 mg of CC/capsule) or placebo (500 mg maltodextrin/capsule) twice daily for 12 weeks. Cognitive function and mood status were assessed at baseline, 6th week, and 12th week using validated neuropsychological tests. Blood biochemical profiles and biomarkers were measured at baseline and 12th week. Two-way mixed analysis of variance (ANOVA) analysis showed significant improvements in mini mental state examination (MMSE) (partial η^2^ = 0.150, *p* = 0.049), tension (partial η^2^ = 0.191, *p* = 0.018), total mood disturbance (partial η^2^ = 0.171, *p* = 0.028) and malondialdehyde (MDA) (partial η^2^ = 0.097, *p* = 0.047) following CC supplementation. In conclusion, 12 weeks CC supplementation potentially improved global cognition, tension, total mood disturbance, and oxidative stress among older adults with MCI. Larger sample size and longer period of intervention with incorporation of metabolomic approach should be conducted to further investigate the underlying mechanism of CC supplementation in neuroprotection.

## 1. Introduction

Aging has become a global issue. The number of people over 60 years old is expected to increase from 507.95 million in 2015 to 1293.7 million by 2050 [1]. Population aging has been fastest in South-East Asia including Malaysia. The percentage of the population aged 65 years and above almost doubled from 6% in 1990 to 11% in 2019 in South-East Asia [2]. In Malaysia, a similar scenario can be observed in which the aging population has risen from 5.6% to 10.7% of the total population between the year 1991 to 2020 [3,4]. The recent World Health Organization report, dementia has been diagnosed among 50 million people globally, with approximately 60% living in low- and middle-income countries. The total number of people with dementia is estimated to reach 82 million in 2030 and 152 million in 2050 [5]. Particularly in Malaysia, the prevalence of dementia in Malaysia is estimated at 0.126% and 0.454% in 2020 and 2050, respectively [3] and the prevalence of mild cognitive impairment is 16% [6].

Mild cognitive impairment (MCI) is an etiologically heterogeneous syndrome characterized by memory performance below the age norm and represents a transitional state between normal aging and dementia disorders. Globally, efforts have been taken to explore neuroprotective effects of nutraceutical products among the aging population aiming for the prevention of neurodegenerative diseases such as dementia through food-based recommendations [7,8,9,10]. Asian in particular is rich in traditional herbs and vegetables potentially use as nutraceutical. Several studies have proven beneficial effects of nutraceutical products on the cognitive function of older adults. These included indigenous herbs rich in polyphenols and antioxidants such as *Persicaria minor* aqueous extract supplementation [11], *Ginkgo biloba* [12,13] and *Centella asiatica* [14]. In particular, for example, an Asian herbs, *Persicaria minor* supplementation for 24 weeks promoted both attention and mood among older adults with mild cognitive impairment [11,15]. Association between dietary nutrients and cognitive function among older adults have been reported elsewhere [16,17,18,19]. Nevertheless, the evidence from randomized controlled trials is limited to justify the recommendation of nutraceutical products are useful to prevent neurodegeneration and cognitive impairment [5,19]. Therefore, more efforts should be conducted to determine the efficacy of the nutraceutical products.

A traditional vegetable, *Cosmos caudatus* (CC) or locally known as *ulam raja* (Kings of *ulam*) is an annual plant in the genus *Cosmos* and it is widely distributed in South East Asia including Indonesia, Thailand, and Malaysia [20]. It showed the highest total phenolic content as compared to nine other common Malaysian traditional vegetables [21]. CC exhibits strong antioxidant properties and may have the ability to prevent neurodegenerative diseases such as dementia [20,22,23]. One of its beneficial based on the medicinal properties are as antidiabetic, anti-obesity, antimicrobial, antihypertensive, and anticancer. However, it is yet to be determined on its potential in dementia prevention [20]. On the other hand, the phytochemicals of CC are known for their free radical scavenging properties that may help to reduce the oxidative stress and lipid peroxidation in neuronal membrane [22,24]. Flavonoid such as quercetin which can be found in CC which potentially improved memory deficits and cognitive impairment in a mice model [25]. In fact, CC also contains other flavonoids such as catechin, quercetin, proanthocyanidin, and rutin which could pass across the blood–brain barrier and might be able to protect against cognitive decline [20]. However, none of the studies examined the effects of CC on cognitive function, mood status, biochemical profiles, as well as biomarkers among older adults with mild cognitive impairment (MCI).

To address the research gap, a randomized, double-blind, placebo-controlled clinical trial was conducted to investigate the effects of 12 weeks of *CC* supplement on cognitive function, mood status, blood biochemical profiles and biomarkers among older adults with MCI. We hypothesized that 12 weeks of CC supplement has the ability to improve cognitive function, mood status, and blood biomarkers among older adults with MCI.

## 2. Materials and Methods

### 2.1. Study Design and Subject’s Selection

A 12-week randomized, double blind, placebo-controlled trial was designed to investigate the effects of CC supplementation on brain function including cognitive function, mood status and biochemical indices among older adults with MCI. The study protocol was approved by the Medical Research Ethics Committee Universiti Kebangsaan Malaysia with code NN-2019-137 and written informed consent was obtained from all the subjects prior to data collection. This study was also registered under the International Standard Randomized Controlled Trial Number (ISRCTN) Registry (ISRCTN16793907) and conducted in accordance with Good Clinical Practice Guidelines and the ethical principles of the Declaration of Helsinki 1964.

Calculation for sample size was determined by using the Randomized Controlled Trials formula proposed by Chan (2003) [26]. Regarding the study of Gschwind et al. (2017), the mean difference of cognitive test score between supplement and control groups (12) with pooled standard deviation (17) were substituted into the formula [13]. The calculated sample size was 24 per group after consideration of 30% drop out rate. The sample size calculation as shown below:
      n1 = n2 = (C/δ^2^) + 2
         *n* = C/(μ2 − μ1/σ)^2^+ 2
       *n* = 7.9/(12/17.4)^2^ + 2
*n* = 18
where *n* = sample size

δ = μ2 − μ1/σ

μ1 − μ2 = mean difference between the intervention and control group

after the intervention [13]

σ = the standard deviation of the intervention group [13]

C = constant: 7.9 (80% power and 95% confidence interval)

With consideration that the dropout rate of subjects is 30%, the sample size would be 18 + 6 = 24 subjects per arm.

This study involved older adults with mild cognitive impairment aged between 60 to 75 years living in the Klang Valley, a central of Malaysia. Subjects were screened and recruited via poster advertisements on social media such as WhatsApp and Facebook. Open health check-ups related to the screening were held at the Centre for Healthy Ageing and Wellness (H-Care) to ensure willingness of the potential subjects to participate into the study. Subjects were allocated based on a simple randomization method using a computer-generated software according to gender run by the investigator with no involvement in the clinical trial. All the study fieldworkers and subjects were kept blinded to the group assignment, study product distribution, and trial findings. The supplement manufacturer unblinded the study label after the data analysis was completed. The inclusion criteria of the study is older adults aged 60 to 75 years old with MCI based on Petersen (2014) criteria [27] and body mass index must be within 20–30 kg/m^2^. Older adults with self-reported neurodegenerative diseases, smokers, regular consumption of traditional herbs or nutraceutical products for the past 6 months, depressive symptoms (score > 5 in Geriatric Depression Scale), contract serious comorbidities such as renal and kidney failure (based on blood analysis report), undergo hormone therapy and consumption of warfarin medication were excluded from the study.

Figure 1 shows the study flow chart. During screening, a total of 200 older adults aged 60 to 75 years were enrolled and assessed for eligibility. A total of 152 older adults were excluded due to several reasons such as not meeting the inclusion criteria (*n* = 121), declined to participate (*n* = 29) and other personal reasons (*n* = 2). Subjects who met the inclusion and exclusion criteria were then randomized into placebo and CC supplement groups, with 24 subjects per arm. On the first follow up at the 6th week, there was one dropout from the CC supplement group due to the subject being uninterested in continuing with the study and failure to answer calls; however, there were no dropouts in the placebo group. In conclusion, only 23 subjects of the CC supplement group and 24 placebo subjects completed the study at the end of the 12-week study (*n* = 47).

### 2.2. Study Product and Dosage

A finished product in the form of a capsule containing 250 mg of CC powder and 250 mg of maltodextrin was produced. This product was developed by the Institute Bioproduct Development (IBD), Universiti Teknologi Malaysia. On a daily basis, two capsules of the supplement were taken by the subjects either during breakfast, lunch or dinner. Each capsule can be orally taken during alternate main mealtime. On the other hand, the placebo used in this study was a 500 mg of maltodextrin. Both capsules were sensory identical and packed in the bottles labelled as either A or B by the manufacturer. We referred to the available toxicity study by Mohamed et al. (2013) which was tested among Wistar rats for acute oral toxicity and it was safe to consume up to 500 mg/kg body weight [28]. In an in vivo animal study examining the antioxidant effects of CC supplement among 30 mice (divided into 5 different doses), findings showed that 100 mg/kg of CC aqueous extract have the potential to increase superoxide dismutase (SOD) and catalase (antioxidant biomarkers) levels after 21 days of treatment [29]. Our dosage (500 mg of CC/day) also exceeded the in vivo animal therapeutic dosage as we hypothesised that it might give the same antioxidant effect in humans. The heavy metals and microbial analysis were conducted and it was reported that composition of heavy metals and microbial was below the toxicity level. The nurition composition of CC supplement and placebo is shown in Table 1. Compliance was assessed by performing capsule count at the end of the 6th and 12th week. The compliance rate of this study was 90.4%.

### 2.3. Study Procedures

Data collection was conducted three times starting from baseline, 6th week and 12th week of supplementation. The data that was taken during that period included details such as subjects’ sociodemographic information, anthropometric measurements (body mass index), vital signs (pulse rate and blood pressure), dietary intake information using validated dietary history questionnaire (DHQ) [30] and neuropsychological tests for every visit. In addition, the blood biochemical profiles and biomarkers were measured during baseline and the 12th week only. A total of five trained fieldworkers participated in the data collection procedures. Those five fieldworkers were assigned to collect the same subjects’ information and conduct the respective cognitive tests from baseline to the 12th week of the intervention to avoid inconsistent results. The data collection was one-to-one and face to face interview. Each neuropsychological tests took 15 to 20 min to complete. Same sets of validated neuropsychological tests were used throughout the study. No pre-testing of the questionnaire was conducted as validated questionnaires were used for all parameters. The outcomes of the study was shown in Table 2.

#### 2.3.1. Cognitive Function Assessment

A series of neuropsychological tests such as Mini-Mental State of Examination [31], Digit Span [32], Rey Auditory Verbal Learning Test [33], Digit Symbol [32], and Visual Reproduction [32] were utilized to assess global cognitive function, working memory, psychomotor speed and visual-spatial memory of the subjects. Their mood status was assessed using the validated Profile of Mood State (POMS) questionnaire [34].

#### 2.3.2. Blood Biochemical Profile and Blood Biomarkers Tests

At baseline and during the 12th week follow up session, subjects were asked to fast overnight for at least 10 h for blood sample collecting purpose. Peripheral venous blood samples were drawn by a trained phlebotomist. A total of 20 mL blood was collected into tubes and immediately stored in an ice box for delivery. All the basic biochemical profile analysis such as fasting blood sugar, lipid profile, liver function test and renal profile were analysed at the medical laboratory Pathlab Malaysia Sdn Bhd, Selangor, Malaysia. The serum was centrifuged and stored under −80 °C for one month before biomarker analysis was carried out using commercial ELISA kits at Bioserasi and toxicology laboratory at the Faculty of Health Sciences, Universiti Kebangsaan Malaysia, Kuala Lumpur. The oxidative stress biomarkers (malondialdehyde, MDA), inflammatory biomarkers (inducible nitric oxide synthase, iNOS and cyclooxygenase-2, COX-2), antioxidant biomarkers (superoxide dismutase, SOD and glutathione, GSH) and brain derived neurotrophic factor (BDNF) were measured in this study using commercial ELISA kits (Elabscience, Houstan, TX, USA). This ELISA kit uses Competitive-ELISA as a method. The standard concentrations of was stated in the kits’ manual. The optical density (OD) was measured spectrophotometrically at a wavelength of 450 nm. The concentrations of all the biomarkers in the duplicate samples were determined by comparing the OD of the samples to the standard curve.

### 2.4. Statistical Analysis

Normality of data was analysed by using the Shapiro-Wilk test and significant value *p* > 0.05 which indicates normal distribution. The mean differences of the baseline data between the supplement and placebo group were analysed using the Independent-*t* test for continuous parameters. On the other hand, Chi-Square was selected for the analysis of categorical variables such as gender or race. Furthermore, the interaction effects were analyzed using a two-way repeated measure analysis of variance (ANOVA) that was adjusted for age, education years, body mass index, physical activity, MMSE, energy intake, vitamin A and C after the Bonferroni adjustment. The covariates and confounding factors for the repeated measure analysis were selected from the variables which may have contributed possible confounding effects on the outcome of measurements. Besides that, the percentage of mean change for each interval (baseline to 6th week and baseline to 12th week) was calculated and presented as a line chart. Percentage of mean change for each subject in both the groups was calculated using the formula; [(score at 6th or 12th week-score of baseline)/score of baseline] × 100%. The mean difference of the percentage mean change in both the groups was analyzed using the Independent-*t*-test.

## 3. Results

### 3.1. Subjects’ Characteristics

Table 3 shows a total of 48 subjects involved in this study with mean age of 65.11 ± 4.05 years. These subjects were randomized into CC supplement group (65.83 ± 4.35 years old) and placebo group (64.42 ± 3.71 years old). Majority of the subjects were women (66.7%), Malay (60.4%), married (77.1%), received secondary education (61.7%) and with a mean household income of RM1991.57 ± 844.94 (USD 475 ± 201). Based on the subjects’ self-reported medical condition, 35.4% of the subjects diagnosed with hypertension, 22.9%, 31.3% and 4.2% of them diagnosed with diabetes, hyperlipidemia and other diseases such as gout and gastroesophageal reflux disease, respectively. The sociodemographic and self-reported medical condition were not statistically significant between both groups at baseline (*p* > 0.05).

### 3.2. Cognitive Function and Mood Status

Table 4 demonstrates the intervention effects after controlling for confounding factors on cognitive functions between CC supplement group and placebo group. Mini-Mental State Examination (MMSE) showed significant intervention effect (*p* = 0.049, partial η^2^ = 0.150, power = 0.586). Both groups showed increment in MMSE mean scores, however, CC supplement group had significant percentage of mean change on the 6th and 12th week of intervention as compared to the placebo group (*p* < 0.05) using the independent-*t* test (Figure 2). Although there were no significant treatment × time effects from the Digit Span test among the CC supplement and placebo groups, independent-*t* test showed that the percentage mean change in the Digit Span test was significantly higher in the CC supplement group (20.6%) than the placebo group (8.32%) on the 6th week of intervention (*p* < 0.05).

A significant intervention effect was observed in tension (*p* = 0.018, partial η^2^ = 0.191, power = 0.733) and total mood disturbance (*p* = 0.028, partial η^2^ = 0.171, power = 0.672). The percentage of mean change for tension and total mood disturbance showed statistically significant improvement in CC supplement as compared to placebo groups at 6th and 12th week (Figure 2).

### 3.3. Biochemical Profiles and Biomarker Analysis

All blood biochemical profiles did not show any treatment × time effects, and there was also no significant difference in percentage mean change between the CC supplement and placebo groups at any time points (*p* > 0.05) which is shown in Appendix A. Subsequently, malondialdehyde (MDA) showed significant intervention effects after 12 weeks of intervention (*p* = 0.047, partial η^2^ = 0.097, power = 0.516) (Table 5). The mean concentration of MDA decreased from 350.14 ± 161.08 to 313.32 ± 153.35 ng/mL (−10.52%) in the CC supplement group. Independent-*t* test also showed the percentage mean change in MDA was significantly decreased from baseline to the 12th week in CC supplement as compared to placebo group (*p* < 0.05). Although there was no significant group, time and intervention effects in glutathione (GSH) between the CC supplement and placebo group, the percentage mean change in GSH was significantly higher in the CC supplement group (+8.7%) as compared to the placebo group (+1.04%) (*p* < 0.05) (Figure 3).

## 4. Discussion

Our study has successfully found that 12 weeks of CC supplementation has the ability to improve global cognitive function as assessed using MMSE. Both CC supplement and placebo groups have improved in these cognitive domains, however, the percentage mean change of CC supplement group was significantly higher than the placebo group. This phenomenon might due to the placebo effect, where placebo subjects expected that the supplements, they received were effective [35]. We are cautious while interpreting the significant result, it showed small effect size, however, it should be recruited more samples to improve the significance of the result in future study. We reasoned the significant improvement as CC supplement contains high flavonoids content such as quercetin, catechin, epicatechin and proanthocyanidins which have a potential involvement in neuroprotection pathway [20,36]. The bioactive compounds such as quercetin and quercitrin of this supplement were higher than the previous proven traditional vegetable *Persicaria minor* supplement in neuroprotection among older adults with MCI [11]. However, the flavonoid contents (excluding quercetin and quercitrin) in CC supplement remain unknown.

In addition, we believe that the cognitive improvement might also cause by the possible synergistic effects between antioxidants and flavonoids activities in CC supplement. It has been reported that certain flavonoids could cross the blood–brain barrier (BBB) in vitro [37,38,39]. The past studies relating to in vivo and in vitro have indicated that flavonoids in herbal extracts that are similar with the CC supplement are competent enough to be absorbed following an oral administration, passing through the blood–brain barrier, and the absorption would lead to different impacts on the central nervous system [38]. Flavonoids such as quercetin might exert antioxidant properties and biochemical impacts with regards to many oxidative stress-induced ailments for the elderly [40,41]. As per emerging evidence, flavonoids have been reported to offer protection against degeneration and neural injuries in dementia and Alzheimer’s diseases [42,43]. However, the types of flavonoids in CC supplement (excluding quercetin) and the bioavailability of CC’s flavonoids in human body is yet to be investigated in future.

Several pathways have been recommended by means of which quercetin and quercitrin in CC supplement might impact cognitive functioning. For instance, in vitro research indicates that quercetin is an effective antioxidant and might safeguard the neuronal cells from neurotoxicity related to oxidative stress [44]. Furthermore, past in vitro studies have pointed out that quercetin is an adenosine antagonist, as it might improve the cognitive functioning and reduce physical and cognitive fatigues [45]. In addition, the quercitrin was found to be involved in the dilution of brain oxidative stress, clampdown of inflammation, and the improvement of neurotransmitter dysfunction [44,46,47]. Flavonoids such as quercetin can deter the mitogen-activated protein kinase (MAPK) signaling pathway by inhibiting the expression of inflammatory protein and cytokine generation [48]. The flavonoids (excluding quercetin and quercitrin) might be beneficial in cognitive functioning, it should be identified their mechanism of action in future.

Additionally, our study shows that there is an improvement in tension and total mood disturbance in CC supplement group as compared to the placebo group. We reasoned that the positive effect of mood might be due to the presence of quercetin and quercitrin in CC supplement. This findings are supported by Yahya et al. (2017) and Lau et al. (2020) in which both studies have reported that the structural formula pertaining to *Polygonum minus* aqueous extract possessing two flavonoids (quercitrin and quercetin) have anti-anxiety and anti-depressant characteristics [11,15]. Likewise, Udani (2013) noted an important enhancement in depression, tension, and anger of POMS following the supplementation of *Superulam*, a mixture of *ulam* extracts which contains CC supplement [49]. When *Nigella sativa* L., also called as black cumin, that also contains quercetin, was supplemented for four weeks, the subject’s mood and anxiety would improve and decrease respectively due to the herbs’ potent anticholinesterase and antioxidant activities [50]. We also reasoned the promising results on mood might be because of the possible synergistic effects between the antioxidants and flavonoids in CC supplement. As stated by Daramola (2018), when natural antioxidant in plants is combined with other antioxidants, it could result in additive and synergistic impacts [51]. However, the underlying mechanisms of CC supplementation on mood is yet to be discovered.

Interestingly, there is a significant decrease in malondialdehyde (MDA) levels in CC supplement group as compared to the placebo group after 12 weeks of supplementation. This phenomenon might be due to the presence of antioxidants and flavonoids in CC supplement are believed to be crucial in stopping the creation of free radicals as it is able to curb few lethal actions of reactive oxygen species (ROS) on DNA, lipids, and proteins [52,53]. The flavonoids in CC supplement also possess the abilities to avert DNA oxidative damage, satiate lipid peroxidation, and rummage reactive oxygen species (ROS) such as hydrogen peroxide, superoxide, and hydroxyl radicals [54]. In addition, we also reasoned that antioxidants and flavonoids in CC supplement has potential to reduce the autoxidation pathway by preventing the formation of free radicals. The flavonoids in CC supplement such as quercetin, catechin, and epicatechin might also possibly aid the species that tend to initiate peroxidation, quench superoxide radicals, and disrupt the autoxidative chain reaction, and inhibit the formation of peroxides [20,55,56].

On the other hand, we have also found that there is a significant increase in the percentage of mean change of serum glutathione (GSH) levels in CC supplement group as compared to the placebo group after 12 weeks of supplementation. Glutathione is an antioxidant that protects against cell damage caused by ROS, including peroxides, lipid peroxides, heavy metals and free radicals [57,58,59]. This could be due to the role of flavonoids in CC that assist the increment of glutathione level in which further scavenged ROS through enzymatic as well as non-enzymatic reactions. The mechanism possibly via the bioavailability of glutathione or glutathione precursors presence that help to boost glutathione levels in the human body [60]. However, the exact amount of glutathione in CC supplement and the bioavailability of glutathione in human body are yet to be investigated.

The possible reasons for our insignificant results in other biochemical profiles and biomarkers may be due to the low bioavailability of flavonoids that reduces the antioxidant and anti-inflammatory capacities, as well as the short duration of the study to improve these biomarkers outcomes. Furthermore, we also did not investigate on the bioavailability of flavonoids from CC supplement in human models. Undoubtedly, there are also other possible risk factors that could affect the biochemical profiles and biomarkers such as stress, social activity involvement, dietary practices, and physical activity that should be considered and measured in future studies [61,62,63,64].

The scientific evidence from this study has the potential to trigger health promotions of CC consumption that could also consequently lead to the population’s healthier lifestyle, improved quality of life, a further reduced risk of contracting neurodegenerative diseases and lowering healthcare costs related to the disease burden [65]. Furthermore, it would also create a “Knowledge-Economy”, of which knowledge would lead to plantations and agricultural industry related to CC plant, thus increase the income of the nation, and further reduce poverty.

The primary strength of our study is the study design of randomized, double-blind, placebo-controlled trial to investigate the effects of CC supplement on cognitive status, biomarkers, health parameters and mood status. This study also focused on older adults with mild cognitive impairment (MCI) who have higher risk of contracting Alzheimer’s disease. The main limitation of this study the sample size is small and the duration of the study is considered short as some of the neuropsychological tests and biochemical profile’s results were not significantly improved. The mood status that was assessed using self-reported questionnaire is less sensitive for neurological disorders, such as cognitive impairment individuals [66]. The flavonoids other than quercetin and quercitrin were not quantified in this supplement. Future studies would benefit from using a larger and more diverse sample size to further explore the efficacy of CC supplements to generate further statistically significant results on cognitive function, biochemical profiles and mood status in a longer period (i.e., 6 months or 12 months). The full bioactive compound profiles should be identified and quantified using liquid chromatography–mass spectrometry method. The metabolomic approach is also recommended to further determine the efficacy of CC to gauge a better understanding of its cognitive decline preventive properties among older adults with MCI [67,68]. Simultaneously, the serum quercetin and other potential flavonoids are suggested to be included in future studies to have a better comprehension on the bioavailability of flavonoids in human body after consuming the CC supplements.

## 5. Conclusions

*Cosmos caudatus* supplement might potentially reduce lipid peroxidation using malondialdehyde biomarkers and increase serum glutathione level. It might also have the ability to improve global cognitive function using neuropsychological test, reduce tension and total mood disturbance among older adults with MCI. The results must be considered preliminary until such effects can be studied in a longer clinical trial and larger sample size to elucidate the neuroprotective effects of CC supplement. The CC supplement reportedly has no harmful effects based on the vital signs, liver function test and renal profile, and there are no serious adverse events were reported after 12 weeks.

## 6. Patents

The patent has been filed with the code: PI2020005611.

## Figures and Tables

**Figure 1 nutrients-13-00434-f001:**
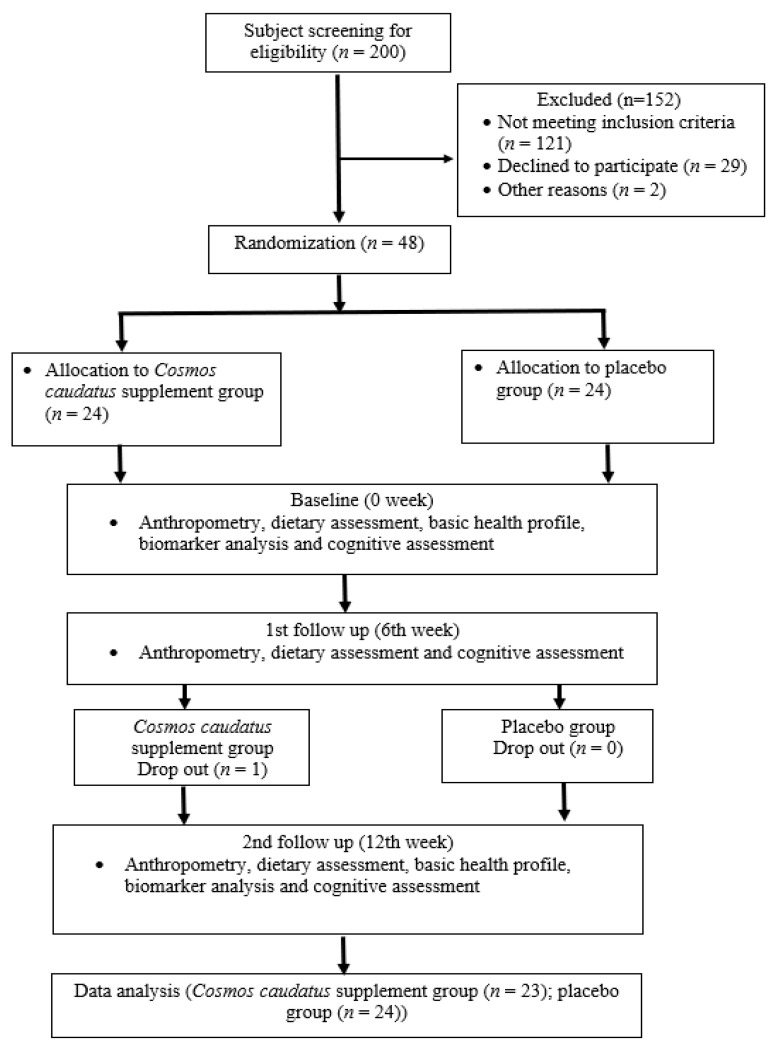
Consort study flow chart.

**Figure 2 nutrients-13-00434-f002:**
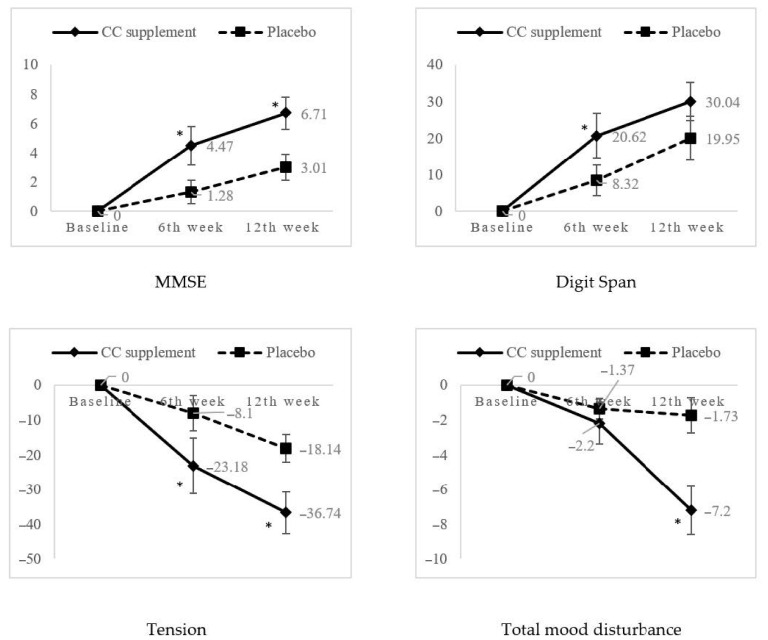
Percentage mean change of significant cognitive and mood parameters from baseline to 6th week and 12th week follow-ups. * Significant at *p* < 0.05 using independent *t*-test.

**Figure 3 nutrients-13-00434-f003:**
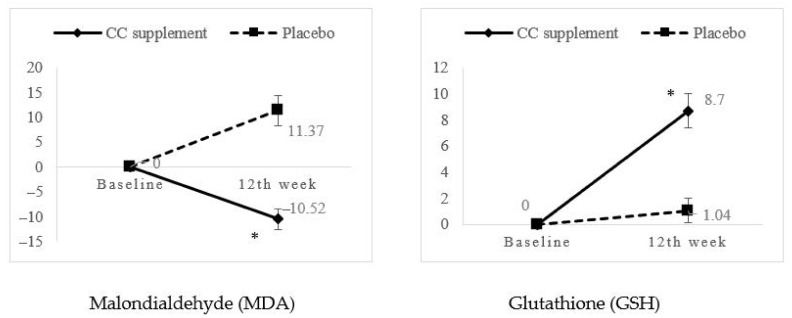
Percentage mean change of significant biomarkers from baseline to 6th week and 12th week follow-ups. * Significant at *p* < 0.05 using independent *t*-test.

**Table 1 nutrients-13-00434-t001:** Nutrient composition of *Cosmos caudatus* (CC) supplement and placebo (per 100 g).

Nutrients	CC Supplement	Placebo
Energy (kcal)	284 ± 4	376 ± 3
Carbohydrate (g)	47.35 ± 0.92	94.05 ± 0.64
Protein (g)	20.30 ± 0.42	0
Fat (g)	1.40 ± 0.14	0
Ash content (g)	26.70 ± 1.27	0.05 ± 0.07
Moisture content (g)	4.25 ± 0.49	5.85 ± 0.50
Vitamin A Retinol (mg)	0	0
Vitamin C (mg)	100.50 ± 2.12	0
Vitamin E Alpha-Tocopherol (mg)	2.10 ± 0.42	0
Calcium (mg)	2255.00 ± 7.07	9.06 ± 0.54
Iron (mg)	7.69 ± 0.16	0.62 ± 0.13
Potassium (mg)	9610 ± 466.69	15.00 ± 0.71
Sodium (mg)	71.60 ± 5.52	69.15 ± 2.62
Zinc (mg)	2.77 ± 0.52	0.70 ± 0.14
Total Dietary Fibre (g)	7.80 ± 2.97	1.8 ± 0.14
Total phenolic content (mg Gallic acid equivalent)	1482 ± 101	3.56 ± 0.10
DPPH (mmol Trolox equivalent)	330.86 ± 9.48	4.57 ± 2.01
FRAP (mmol Trolox equivalent)	393.57 ± 9.78	1.45 ± 1.17
Quercetin (%*w/w*)	0.9	NA
Quercitrin (%*w/w*)	1.0	NA

The data are nutritional information of CC supplement and placebo per 100 g, where subjects consumed two capsules per day (500 mg/day) for the intervention. Abbreviation: DPPH: 2,2-diphenyl-1-picrylhydrazyl; FRAP: Ferric Reducing Antioxidant Power.

**Table 2 nutrients-13-00434-t002:** Outcomes of the study.

Outcomes	Neuropsychological Assessment
Primary outcomes	
MMSE	Global cognitive function
Digit Span	Attention and working memory
RAVLT (immediate and delayed recall)	Verbal immediate memory
VR (immediate and delayed recall)	Visuo–spatial function
Digit symbol substitution	Psychomotor speed
POMS	Tension, depression, anger, fatigue, esteem-related effect, vigor, and confusion
Secondary outcomes	
Biomarkers	BDNF, MDA, iNOS, COX-2, SOD, GSH
Biochemical profiles	Fasting blood sugar, lipid profile, liver function test, renal function test

Abbreviation: BDNF: Brain derived neurotrophic factor; COX-2: Cyclooxygenase-2; GSH: Glutathione; iNOS: Inducible nitric oxide synthase; MDA: Malondialdehyde; MMSE: Mini-mental State Examination; POMS: Profile of Mood State; RAVLT: Rey Auditory Verbal Learning Test; SOD: Superoxide dismutase; VR: Visual reproduction.

**Table 3 nutrients-13-00434-t003:** Baseline sociodemography information and self-reported medical condition between CC supplement and placebo group subjects [presented as mean ± standard deviation or *n*(%)].

Parameter	CC Supplement (*n* = 24)	Placebo (*n* = 24)	Total (*n* = 48)	*p*-Value
Age ^1^	65.83 ± 4.35	64.42 ± 3.71	65.11 ± 4.05	0.237
Gender ^2^				0.917
Male	8 (33.33)	8 (33.3)	16 (33.3)	
Female	16 (66.7)	16 (66.7)	32 (66.7)	
Ethnicity ^2^				0.483
Malay	16 (66.7)	13 (54.2)	29 (60.4)	
Chinese	6 (25.0)	10 (41.7)	16 (33.3)	
Indian	2 (8.3)	1 (4.2)	3 (6.3)	
Formal education (years) ^1^	11.39 ± 2.39	10.17 ± 3.20	10.77 ± 2.87	0.145
Education level ^2^				0.184
Primary school	1 (4.2)	3 (12.5)	4 (8.3)	
Secondary school	13 (54.2)	17 (70.8)	30 (62.5)	
Diploma/Certificate	9 (37.5)	3 (12.5)	12 (25.0)	
Degree	1 (4.2)	1 (4.2)	2 (4.2)	
Marital status ^2^				0.123
Single	1 (4.2)	3 (12.5)	4 (8.3)	
Married	22 (91.7)	15 (62.5)	37 (77.1)	
Divorce	0 (0)	2 (8.3)	2 (4.2)	
Widow/widower	1 (4.2)	4 (16.7)	5 (10.4)	
Household income (RM) ^1^	2021.83 ± 904.81	1962.58 ± 801.86	1991.57 ± 844.94	0.813
Hypertension ^2^				0.159
Yes	6 (25.0)	11 (45.8)	17 (35.4)	
No	18 (75.0)	13 (54.2)	31 (64.6)	
Diabetes ^2^				0.671
Yes	6 (25.0)	5 (20.8)	11 (22.9)	
No	18 (75.0)	19 (79.2)	37 (77.1)	
Hyperlipidaemia ^2^				0.587
Yes	7 (29.2)	8 (33.3)	15 (31.3)	
No	17 (70.8)	16 (66.7)	33 (68.7)	
Others ^2^				0.975
Yes	1 (4.2)	1 (4.2)	2 (4.2)	
No	23 (95.8)	23 (95.8)	46 (95.8)	
Physical activity ^2^				0.591
Everyday	1 (4.2)	0 (0)	1 (2.1)	
3–5 times per week	7 (29.2)	4 (16.7)	11 (22.9)	
1–2 times per week	8 (33.3)	11 (45.8)	19 (39.6)	
None	8 (33.3)	9 (37.5)	17 (35.4)	
Body mass index (kg/m^2^)	25.67 ± 3.02	25.72 ± 2.29	25.70 ± 2.64	0.952

^1^ Independent-*t* test, not significant at *p* > 0.05; ^2^ Cross tabs Chi-square test, not significant at *p* > 0.05; N/A: Not applicable.

**Table 4 nutrients-13-00434-t004:** Intervention effect of cognitive function and mood state from baseline to 12th week.

	CC Supplement (*n* = 23)	Placebo (*n* = 24)	Treatment × Time Effect		
			*p*	Partial Eta Squared	Power
Mini-mental State Examination (MMSE)					
Baseline	27.09 ± 1.38	26.58 ± 1.35	0.049 *	0.150	0.586
6th week	28.30 ± 0.70	26.92 ± 1.02			
12th week	28.91 ± 0.95	27.38 ± 1.28			
Digit Span					
Baseline	8.39 ± 1.23	7.92 ± 1.28	0.466	0.040	0.173
6th week	10.09 ± 1.41	8.58 ± 2.34			
12th week	10.91 ± 1.70	9.50 ± 2.59			
RAVLT (Immediate Recall)					
Baseline	6.30 ± 1.02	6.29 ± 1.12	0.058	0.143	0.560
6th week	10.22 ± 1.28	8.96 ± 1.76			
12th week	11.26 ± 2.18	10.75 ± 2.29			
RAVLT (Delayed recall)					
Baseline	5.74 ± 0.92	5.63 ± 1.24	0.070	0.068	0.529
6th week	9.39 ± 1.37	7.83 ± 1.49			
12th week	9.96 ± 2.65	9.58 ± 3.02			
Digit Symbol					
Baseline	8.87 ± 1.71	8.00 ± 2.63	0.264	0.069	0.278
6th week	10.04 ± 2.01	8.88 ± 2.59			
12th week	10.48 ± 1.97	9.88 ± 2.89			
Visual Reproduction (Immediate recall)					
Baseline	32.22 ± 4.62	29.50 ± 6.45	0.212	0.080	0.320
6th week	34.83 ± 3.30	29.54 ± 7.25			
12th week	35.13 ± 3.94	30.88 ± 7.06			
Visual reproduction (delayed recall)					
Baseline	31.61 ± 6.06	27.83 ± 7.98	0.242	0.074	0.295
6th week	35.13 ± 4.35	28.88 ± 7.85			
12th week	35.30 ± 3.40	29.88 ± 8.73			
Tension					
Baseline	5.09 ± 3.46	6.67 ± 3.95	0.018 *	0.191	0.733
6th week	3.91 ± 2.04	6.13 ± 2.53			
12th week	3.22 ± 1.65	5.46 ± 2.25			
Anger					
Baseline	1.47 ± 0.92	2.58 ± 1.16	0.139	0.099	0.401
6th week	1.13 ± 1.58	3.21 ± 1.83			
12th week	1.09 ± 0.64	3.08 ± 2.39			
Fatigue					
Baseline	3.65 ± 2.53	5.17 ± 2.35	0.811	0.011	0.081
6th week	4.17 ± 1.87	5.33 ± 2.91			
12th week	4.00 ± 1.57	5.54 ± 1.25			
Depression					
Baseline	1.70 ± 0.36	3.00 ± 1.22	0.921	0.004	0.062
6th week	1.39 ± 0.37	2.38 ± 2.20			
12th week	1.34 ± 0.77	2.17 ± 1.93			
Esteem-related effect					
Baseline	16.74 ± 2.88	14.54 ± 3.06	0.149	0.095	0.387
6th week	17.30 ± 1.69	15.21 ± 3.46			
12th week	18.35 ± 1.34	15.00 ±2.83			
Vigor					
Baseline	12.83 ± 3.39	10.25 ± 3.72	0.050	0.243	0.862
6th week	13.30 ± 1.55	10.29 ± 2.16			
12th week	14.35 ± 2.17	9.00 ± 2.11			
Confusion					
Baseline	2.74 ± 1.15	4.13 ± 1.84	0.983	0.001	0.052
6th week	3.22 ± 2.15	3.88 ± 2.86			
12th week	2.00 ± 1.45	2.83 ± 1.34			
Total mood disturbance					
Baseline	85.09 ± 10.58	96.75 ± 12.74	0.028 *	0.171	0.672
6th week	83.22 ± 6.18	95.42 ± 12.04			
12th week	78.96 ± 5.06	95.08 ± 9.08			

* Significance at *p* < 0.05. Controlled for age, body mass index, physical activity, energy intake, vitamin A and C.

**Table 5 nutrients-13-00434-t005:** Intervention effect of biomarkers from baseline to 12th week.

	CC Supplement (*n* = 23)	Placebo (*n* = 24)	Treatment × Time Effect		
			*p*	Partial Eta Squared	Power
Brain derived neurotrophic factor (BDNF) (pg/mL)					
Baseline	180.80 ± 86.42	180.09 ± 95.78	0.884	0.001	0.052
12th week	194.74 ± 137.49	171.22 ± 113.87			
Inducible nitric oxide synthase (iNOS) (pg/mL)					
Baseline	220.38 ± 66.56	218.99 ± 74.66	0.994	0.000	0.050
12th week	216.34 ± 81.39	219.64 ± 80.67			
Cyclooxygenase-2 (COX-2) (ng/mL)					
Baseline	0.86 ± 0.42	1.08 ± 0.59	0.867	0.001	0.053
12th week	0.78 ± 0.43	1.15 ± 0.70			
Superoxide dismutase (SOD) (pg/mL)					
Baseline	49.13 ± 3.81	49.46 ± 2.64	0.577	0.008	0.085
12th week	51.63 ± 2.04	51.22 ± 3.50			
Malondialdehyde (MDA) (ng/mL)					
Baseline	350.14 ±161.08	358.21 ± 172.59	0.047 *	0.097	0.516
12th week	313.32 ± 153.35	398.94 ± 181.85			
Glutathione (GSH) (mM)					
Baseline	0.46 ± 0.09	0.48 ± 0.05	0.111	0.064	0.356
12th week	0.50 ± 0.06	0.48 ± 0.07			

* Significance at *p* < 0.05. Controlled for age, physical activity, BMI, energy intake, vitamin A and C.

## Data Availability

The datasets of this study available from the corresponding author on reasonable request.

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
