# Peer review of "Effects of 12 Weeks Cosmos caudatus Supplement among Older Adults with Mild Cognitive Impairment: A Randomized, Double-Blind and Placebo-Controlled Trial"

_nutrients, 2021, doi:10.3390/nu13020434_

Round 1
Reviewer 1 Report
This study has successfully found that 12 weeks of Cosmos caudatus supplementation has the ability to improve global cognitive function and working memory.
Please specify which flavonoids appear in CC, in addition to quercetin, since the Authors write about a high content of flavonoids. Please refer to the different proven properties of specific flavonoids or groups of flavonoids based on their structure and previous research - not only antioxidant (well known) and neurodegenerative.
Please specify which neurodegenerative diseases are being analyzed in this study as this is the definition of a large pool of diseases.
Please clearly state the hypothesis and the aim of the research.
Please indicate potential for future research. What is the measurable benefit of the conducted research?
Line 253: "Statustical" --> "statistical"
Lines 367-376: "We reasoned the significant improvement as CC supplement contains high flavonoids content such as quercetin, catechin, epicatechin and proanthocyanidins which have a potential involvement in neuroprotection pathway [20, 36]." "certain flavonoids could cross the blood-brain barrier (BBB) in vitro [37-39]." -->Please describe in detail the mechanism of action of these phenolic compounds in neurodegenerative diseases. Can these activities be linked to the results obtained?
Author Response
Dear reviewer,
Thank you for your valuable feedback and comments. We appreciate it and already correct based on comments. Have a nice day.

Reviewer 2 Report
Introduction: Please be cautious when using the term "antioxidant" as it can be misleading. Many compounds might act as antioxidant in vitro, but no longer in vitro. You also state that CC contains quercetin - is this correct, or is it a quercetin glycoside?
Methods:
- Please state how volunteers were recruited, and please expand the power calculation (i.e. it should be understandable without having to obtain the reference).
- Could you please state how the cognitive tests were administered?
- Please state your quality control and quality assurance procedures, in particular assay precision and accuracy, number of control samples and range.
- The authors conduct a large number of tests - how did you address the issue of multiple testing?
Results:
- See also above - you are conducting a large number of tests and assign p=0.05 as threshold for significance. While it is technically correct to interpret 0.049 as 'statistically significant', it might be sensible to be a bit more cautious.
- Please include error bars in all plots
- There are two table 4
Discussion:
I disagree with the conclusion, that the study has shown significant effects - given the number of tests conducted, and the high variability, I believe that it is not possible to interpret the results in either direction. Moreover, the differences in intervention and placebo are not restricted to phenolic compound, and that should also be discussed.
Author Response
Dear reviewer,
Thank you for your comments, we appreciate it. It helped us to improve a lot in our manuscript. I hope we answered all the comments. Thank you again and have a nice day.

Round 2
Reviewer 2 Report
The authors have addressed my comments.